# Injury Patterns after Falling down Stairs—High Ratio of Traumatic Brain Injury under Alcohol Influence

**DOI:** 10.3390/jcm11030697

**Published:** 2022-01-28

**Authors:** Jason-Alexander Hörauf, Christoph Nau, Nils Mühlenfeld, René D. Verboket, Ingo Marzi, Philipp Störmann

**Affiliations:** Department of Trauma, Hand and Reconstructive Surgery, Hospital of the Johann Wolfgang Goethe—University Frankfurt am Main, 60590 Frankfurt am Main, Germany; Christoph.Nau@kgu.de (C.N.); nils.muehlenfeld@kgu.de (N.M.); Rene.Verboket@kgu.de (R.D.V.); Ingo.Marzi@kgu.de (I.M.); Philipp.stoermann@kgu.de (P.S.)

**Keywords:** stairs, fall, traumatic brain injury, alcohol, trauma

## Abstract

Falling down a staircase is a common mechanism of injury in patients with severe trauma, but the effect of varying fall height according to the number of steps on injury patterns in these patients has been little studied. In this retrospective study, prospectively collected data from a Level 1 Trauma Center in Germany were analyzed regarding the injury patterns of patients admitted through the trauma room with suspicion of multiple injuries following a fall down a flight of stairs between January 2016 and December 2019. In total 118 patients were examined which where consecutively included in this study. More than 80% of patients suffered a traumatic brain injury, which increased as a function of the number of stairs fallen. Therefore, the likelihood of intracranial hemorrhage increased with higher numbers of fallen stairs. Fall-associated bony injuries were predominantly to the face, skull and the spine. In addition, there was a high coincidence of staircase falls and alcohol intake. Due to a frequent coincidence of staircase falls and alcohol, the (pre-)clinical neurological assessment is complicated. As the height of the fall increases, severe traumatic brain injury should be anticipated and diagnostics to exclude intracranial hemorrhage and spinal injuries should be performed promptly to ensure the best possible patient outcome.

## 1. Introduction

According to the World Health Organization (WHO), falls are the second most common cause of accidental death from injury worldwide [1]. Due to the height of the fall and the forces acting on the human body, falls can generally be divided into falls at ground level (low-energy trauma) and falls from a greater height (high-energy trauma). Depending on the trauma mechanism, both fall groups are associated with particular injury patterns. Thereby, ground-level falls particularly affect the limbs, followed by the head [2], whereas falls from greater heights are related to injuries of the axial skeleton, such as the spinal column, or pelvic injuries, as well as head injuries [3,4]. Compared to ground level falls and falls from greater heights, stair falls are much less studied. In particular, data on injury patterns and injury severity as a function of the number of stairs fallen are limited.

In recent decades, the number of stair-associated falls has steadily increased, regardless of the patient age [5]. In the course of demographic change, older patients are more frequently affected by stair falls due to their frailty and dwindling muscle strength as well as general multimorbidity. Falls in this age group are especially associated with a greater severity of both trauma and mortality [6,7]. Additionally, falling down stairs also shows a high incidence in the middle-aged population, but these falls are often associated with alcohol consumption [8,9]. Given the local conditions with a well-developed subway network, falls while entering and leaving the stations are frequent. Since falls from stairs occur regardless of age and are often accompanied by a complex pattern of injuries, the socioeconomic burdens caused by long stays in the intensive care unit, prolonged rehabilitation and, in some cases, permanent neurological impairments are high [10].

The aim of the present retrospective study is to examine stair-associated falls for epidemiological characteristics and, in particular, injury patterns and severity in the context of the number of steps fallen.

## 2. Materials and Methods

Prospectively collected data of patients who were admitted to our Level 1 Trauma Center with suspicion of severe trauma following activation of the trauma team depending on the on-scene classification of activation criteria of either a fall below 3 m or a fall over 3 m according to the guidelines of the German Trauma Society (DGU) between January 2016 and December 2019 were evaluated. All clinical data were prospectively taken during the quality documentation of the TraumaRegister of the DGU, Berlin, Germany. The accident mechanism (stair/escalator) was taken from the handover of the prehospital emergency team and, if documented, the preclinically estimated fall height was added in number of steps.

The patients were grouped according to the number of steps fallen. The first group consisted of patients with a documented fall height of 1–5 stairs (low fall), the second group with a fall height of 5–10 stairs (intermediate fall) and the third group with a fall height of more than 10 stairs (high fall). Those patients with unknown fall height and age younger than 18 years were excluded from further analyses (Figure 1).

Basic patient data, including demographics, preclinical and clinical (after trauma room admission) vital parameters (e.g., Glasgow Coma Scale [GCS], systolic blood pressure [SBP], and heart rate [HR]), injury pattern (using different common injury severity assessment tools such as the Abbreviated Injury Scale [AIS] and the trauma scores calculated on the basis of the AIS, such as the Injury Severity Score [ISS]), surgical treatment, serum blood alcohol concentration, pre-existing anticoagulant medication and outcome parameters (length of intensive care unit treatment, length of hospital stay [LOHS], discharge home or to a follow-up rehabilitation facility) were abstracted from the institutional patient charts. Whole-body computed tomography scans, which are performed as the standard for the treatment of patients with potentially severe injuries, were used as the basis for analyzing the injury patterns.

The demographic and clinical characteristics comparing the different fall height groups (low, intermediate, or high) were evaluated using bivariate analysis. The *p*-values for categorial variables were derived from the Chi-square test or two-sided Fisher’s exact test and for continuous variables from Student’s t-test or the Mann–Whitney U test. The significance level was defined as *p* < 0.05 in bilateral testing.

The descriptive and comparative statistical analyses were performed using the IBM Statistical Package for Social Sciences (SPSS) version 25.0 (SPSS Inc., Chicago, IL, USA) for Mac. Values are reported as means ± standard deviations (SDs) for continuous variables and as percentages for categorical variables.

## 3. Results

After excluding those patients with unknown fall height, a total of 118 patients were included for analyses (Figure 1). Patients were divided into three different groups depending on the reported fall height (indicated by the number of steps): 20 patients were assigned to the low fall group (1–5 steps), 40 to the intermediate fall group (6–10 steps), and 58 to the high fall group (>10 steps), respectively. 

The mean age of the patients was 59.5 ± 22.8 years, and 67% of the patients were male. The number of domestic stair falls and outdoor stair falls were distributed equally. Stair falls occurred four times more frequent than escalator falls. On average, patients fell down 11.3 ± 5.6 steps, with patients from the high fall height group falling 16 ± 3.7 steps, which was significantly more steps than those from the other groups. The median Injury Severity Score (ISS) was 10.3 ± 9.8 points (pts.), and the median New Injury Severity Score (NISS) was 13.8 ± 15.5 points, respectively. The ISS increased with increasing fall height. Similar results were seen for the NISS (Table 1).

The mean AIS score for the head was 2.01 ± 1.55 points, for the face 0.33 ± 0.74 points, for the thorax 0.52 ± 1.12 points, for the abdomen 0.25 ± 0.70 points and for the extremities 0.31 ± 0.75 points. The AIS_head_ increased in relation to the height of the fall. Thus, the highest AIS_head_ (2.22 ± 1.61 points) was observed in the high fall group (Table 1).

The Glasgow Coma Scale (GCS) was recorded both at the scene of the accident (GCS_at scene_) and on admission to the trauma room (GCS_in-hospital_). The mean GCS_at scene_ was 11.7 ± 4.3 points, and the mean GCS_in-hospital_ was 12.1 ± 4.7 points. For both GCS values, the comparison between the different groups showed no significant differences (Table 2).

More than 80% (n = 98) of the examined patients sustained a TBI, and this was defined as an occurrence of at least cerebral concussion with a requirement for consecutive 24 h in-hospital monitoring. Similar to the AIS_head_ scores, the incidence of TBI also increased with the number of stairs the patients had fallen (Table 3). 

With respect to intracranial bleeds, subdural hemorrhage (SDH), subarachnoidal hemorrhage (SAH), intracerebral hemorrhage (ICH) and epidural hemorrhage (EDH) were most frequent in the high fall group (Table 3). Of these patients suffering from intracranial hemorrhage, 12 patients (10.2%) required emergency neurosurgical intervention. In 27 patients (22.9%), anticoagulant medication prior to the accident was registered (Table 1).

A total of 56 patients had at least one fracture, with a significantly higher fracture frequency in the high fall group (Table 3). Overall, fractures of the skull and midface were most frequent (mainly fractures of the nose, zygomatic bone and mandible, data not shown). Spine fractures were the second most frequent region, with the cervical and thoracic spine each being affected fourteen times and the lumbar spine being affected ten times. In twelve patients, at least one rib fracture was documented. Among the extremities, the distal radius was the most commonly affected (n = 5). Except for facial fractures, there was no significance of fracture frequencies of the aforementioned regions between the different fall height groups. Excluding operative treatment of intracranial hemorrhage, 29 patients (24.6%) underwent surgical treatment (Table 4). 

The mean length of stay (LOS) in the intensive care unit [ICU] was 2.9 ± 6.4 days. Therefore, an increasing number of fallen steps led to longer ICU stays. The mean length of stay at the hospital (LOSH) was 7.4 ± 9.1 days, with patients from the high fall group requiring the longest in-hospital treatment. After treatment in the hospital, most patients (73.7%) were discharged home. Fifteen patients (12.7%) were transferred directly to a neurological rehabilitation facility, with the largest proportion being from the high fall group. The overall in-hospital mortality was 11% (Table 4).

In 44.9% of the patients, alcohol was detected in the blood (Table 2). Among the 53 patients with positive blood alcohol concentrations (BAC+), the mean BAC was 2.45 ± 0.98 g/dL or 1.99 ± 0.79‰, respectively (Table 5). The fall height of patients under the influence of alcohol and sober patients (BAC−) was comparable. For alcohol-intoxicated patients, both the ISS and NISS showed lower scores, nevertheless, lacking statistical significance. Additionally, BAC+ patients showed lower AIS scores for AIS_head_, AIS_thorax_ and AIS_extremities_. Both the ICU length of stay and LOS in the hospital were comparable between both groups (Table 5).

## 4. Discussion

In the present study, we demonstrated that stair and escalator falls are associated with TBI, which increases with the number of steps fallen. Despite the increasing demographic change, younger patients also, in great part due to alcohol intoxication, frequently suffer from substantial harm when falling down stairs, therefore representing a relevant burden on health care systems.

Since stairs are so widespread in society and falls are frequent, it is important to investigate stair-related injuries in the population. In contrast to low-energy trauma such as ground-level falls or high-energy trauma such as free falls from greater heights, we assume that stair falls represent a special and diverse fall entity due to the complex nature of their mechanism. Whereas the elderly are often injured due to low-energy trauma such as falls at ground-level [11], high-energy trauma, such as motor vehicle collisions or free falls from greater heights occur most frequently in young patients [3,12]. In our recent study, stair falls occurred in all age groups, indicating that stairway falls are not a domain of the elderly. The proportion between indoor and outdoor stair falls in the present study was balanced, and the patients who suffered from domestic falls were older than patients falling down stairs in a public environment. These findings are consistent with a variety of studies. Thus, indoor falls are associated with advanced age and with frailty, medical conditions or long-term medication [13]. On the other hand, younger people in particular are more likely to be affected by outdoor falls due to their higher activity level [14].

The risk of falling is influenced by both intrinsic and extrinsic factors. Extrinsic factors mainly include environmental factors such as uneven and slippery floors or poor lighting conditions [15]. Intrinsic factors relate to the constitution of the individual, such as the balance ability, gait, muscle strength or visual acuity [16]. In different studies it could be shown that the risk of falling could be reduced considerably by preventive measures such as marking stair steps, attaching banisters or installing elevators [15,17]. Thus, Keall et al. showed in a cluster-randomized controlled trial that home modifications consisting of handrails for steps and stairs, high-visibility and slip-resistant edging for steps as well as fixing of lifted edges of carpets and mats resulted in a 26% reduction in injuries attributable to home falls that needed medical treatment [17]. Other studies have shown that balance training and muscle strengthening were the main factors in significantly reducing the risk of falls [18,19]. In order to minimize the number of stair falls in the future, on the one hand preventive measures such as marked stair steps or non-slip surfaces have to be implemented and on the other hand, especially with regard to the aging population, guided training programs for this part of the population to maintain physical fitness into old age have to be offered. Nevertheless, due to the high number of patients in our study who were alcohol-intoxicated, the true benefit of those preventive measures remains elusive.

Due to their staircase falls, about one-quarter of the patients in our collective suffered a subarachnoid hemorrhage, followed by a subdural hemorrhage and intracerebral hemorrhage, with about 20% each. The Abbreviated Injury Scale (AIS), a common scoring system for stratification of trauma severity, was also the highest for head injuries compared to the AIS of the other body regions. Similar to the high incidence of TBI observed in this study, ground-level falls, especially in elderly patients, also show high rates of TBI, with intracranial hemorrhage accounting for 70% of the brain injuries [20]. While stair falls are significantly less common than ground-level falls, a study by Boye et al. assessing the circumstances of falls of the elderly revealed that stair falls were the most common indoor activity leading to TBI in half of the cases [21]. In this context, Hwang and colleagues reported that patients who fall on stairs have about a 3-fold higher risk of suffering from a moderate/severe TBI compared to patients who fell while walking [22]. In line with these findings, we were able to show that, with an increasing number of steps, the incidence of TBI and intracranial bleeding increased. Even though TBIs are a common pattern of injury after stair falls in other studies, in our study, TBIs are particularly prevalent, accounting for more than 80% of cases [5,23]. The high rate of TBI in our study might be due to the fact that all patients included in this study were primarily admitted via our trauma room, indicating that a severe trauma was assumed a priori in the preclinical setting. Therefore, some stair falls that did not result in (severe) TBI may not have been admitted through our trauma room and were therefore excluded for the present data collection.

Consistent with the high rate of TBIs observed in this study, patients in our collective were likely to suffer facial skull fractures, indicating the need for specialized surgeons during the treatment of these patients. Interestingly, fractures of the extremities were quite rare. This is possibly due to the fact that physical strength, reaction time and coordination skills decrease with increasing age [24]. This may lead to insufficient performance of protective movements, such as extending the arms to protect against impact, resulting in direct trauma to the head and torso. Moreover, the relatively low number of fractures of the extremities could be explained by degraded reaction time due to the high rate of alcohol-intoxicated patients.

According to the S3 guidelines of the German Trauma Society (DGU), trauma patients who fall are usually categorized as “falls below 3 m” or “falls above 3 m”, depending on the estimated height of the fall [25]. In clinical routine, to the best of our knowledge, stair falls are also classified according to this classification, with stair falls often assigned to falls greater than 3 m. However, especially in the case of free falls, the body is exposed to significantly stronger compression and deceleration forces during the impact than in the case of a stair fall, potentially resulting in a different spectrum of injuries. Thus, fractures of the lower spine and the pelvis occur significantly more often after a fall from a greater height compared to the injury patterns surveyed in this study [26,27]. Moreover, extremity fractures occurred significantly less frequent in the present study than has been described in the literature following free falls [27]. Additionally, bilateral extremity fractures are also more common and are especially prevalent in suicidal jumps, because the victims sustain feet-first impacts [27]. Moreover, intrathoracic injuries such as (hemato-) pneumothoraces or pulmonary lacerations and intraabdominal injuries such as splenic or hepatic lacerations, which often present after free falls, were also not predominantly seen in our studied collective [4,26]. 

It has been shown that injury severity correlates with fall height in free falls [4,26,27]. In our work, we were able to show that the ISS and NISS, respectively, increased with increasing stair fall height. With regard to the injured body regions, the AIS scores of the head and face in particular increased with increasing stair fall height, suggesting that head impact is the leading consequence of a stair fall. The kinetic energy acting on the body during a fall increases due to acceleration during the fall and is maximum at the moment of impact [28]. Compared to free falls, the acceleration during stair falls is less, resulting in a lower kinetic energy acting on the body, which leads to a markedly different injury pattern. Due to these differences in potential injury patterns, stair falls should be differentiated in addition to falls greater than 3 m (free falls) and less than 3 m (ground-level falls) to more accurately anticipate potential injury consequences and thus alert the appropriate trauma team to ensure the best possible clinical outcome for patients.

As mentioned before, falls recorded in the present study were frequently associated with alcohol consumption. Thus, approximately 45% of the patients had a positive blood alcohol concentration at admission, which is consistent with the current literature that describes a prevalence of alcohol intoxication of 37–51% at the time of the injury [29]. 

Although the patients consuming alcohol in this study fell on average one stair step more than the sober patients, they had a lower ISS and NISS, respectively. The overall lower injury severity was associated with both a shorter length of stay in the ICU and in the hospital in general for alcohol-intoxicated patients. Interestingly, some clinical studies have reported reduced mortality and improved outcomes of acute alcohol-intoxicated patients after TBI [30,31]. These outcome-improving effects of alcohol may be due to its anti-inflammatory properties, which have been shown in numerous studies, both in vitro and in vivo [32,33,34]. In a murine blunt TBI model, oral alcohol gavage prior to traumatic head injury resulted in faster and better neurological recovery of the alcoholized mice, which was accompanied by suppressed pro-inflammatory markers in the brain parenchyma [35]. The effects of alcohol on the further course after stair falls observed in this study may be due to an attenuation of the posttraumatic immune response, which in the presented case is associated with a shortened need for clinical treatment.

## 5. Conclusions

The present study shows that stair falls occur amongst all ages and result in a heterogeneous pattern of injury, with TBIs being the most common. The high coincidence of stair falls and alcohol intoxication observed in this study poses an additional challenge for the treating physicians. Due to the interdisciplinary approach with a high probability for neurosurgical and facial surgery, patients should preferably be treated in hospitals that have such departments. Patients who have suffered a fall down several flights of stairs in particular should therefore not be underestimated in the possible pattern of injury but need the full clinical approach of trauma team activation and polytrauma diagnostic standards, including standardized priority-oriented trauma room management, such as Advanced Trauma Life Support (ATLS).

## 6. Limitations

First, the data analyzed in the present study were obtained from a single hospital, serving as a Level 1 Trauma Center, and included only data from patients presenting via the trauma room. Therefore, patients suffering a staircase fall without suspicion of severe trauma were excluded a priori. Referring to this, the number of total stair falls is likely to be higher and the injury severity may be lower on average. Furthermore, if the number of steps fallen was not known, patients were excluded from this study to better analyze the impact of the height patients fell. To validate and possibly complement the results of this work in future studies, prospective studies should be conducted in which the number of fallen stairs is accurately recorded preclinically or at the scene, respectively.

## Figures and Tables

**Figure 1 jcm-11-00697-f001:**
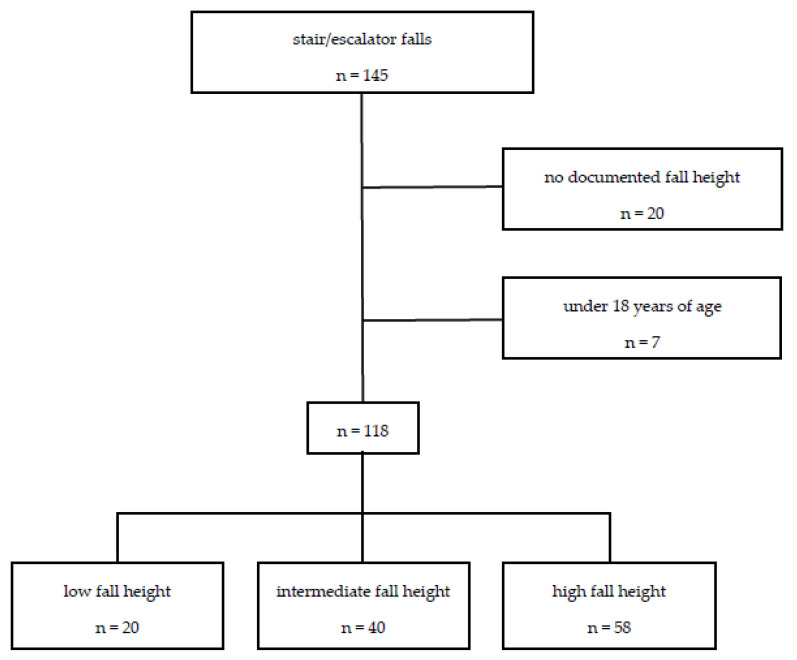
Patients were categorized into three groups based on documented fall height. Fall victims without documented fall height as well as patients under the age of 18 were excluded.

**Table 1 jcm-11-00697-t001:** Demographic data. ^1^

	Total (n = 118)	Low Fall Height (n = 20)	Intermediate Fall Height (n = 40)	High Fall Height (n = 58)	*p*-Value
Fall height (number of steps), mean ± SD	11.3 ± 5.6	3.9 ± 1.1	8.3 ± 1.6	16 ± 3.7	0.000
Age (years), mean ± SD	59.5 ± 22.8	62.9 ± 26.7	60.2 ± 22.7	57.8 ± 21.7	0.672
sex (male)	66.9% (n = 79)	55% (n = 11)	75% (n = 30)	65.5% (n = 38)	0.284
ISS (points), median (IQR)	5.5 (14)	5.5 (8)	5 (11)	9 (20)	0.468
NISS (points), median (IQR)	6 (19)	5.5 (9)	6 (14)	9 (21)	0.537
AIS_head_ (points), mean ± SD	2.01 ± 1.6	1.65 ± 1.4	1.88 ± 1.5	2.22 ± 1.6	0.290
AIS_face_ (points), mean ± SD	0.33 ± 0.7	0.25 ± 0.6	0.20 ± 0.6	0.45 ± 0.9	0.231
AIS_thorax_ (points), mean ± SD	0.52 ± 1.1	0.5 ± 1.1	0.45 ± 1.2	0.57 ± 1.1	0.873
AIS_abdomen_ (points), mean ± SD	0.25 ± 0.7	0.25 ± 0.8	0.18 ± 0.6	0.29 ± 0.7	0.719
AIS_extremities_ (points), mean ± SD	0.31 ± 0.8	0.25 ± 0.6	0.35 ± 0.8	0.31 ± 0.8	0.888
domestic fall	50% (n = 59)	45% (n = 9)	57.5% (n = 23)	46.5% (n = 27)	0.503
stair fall	81.4% (n = 96)	85% (n = 17)	80% (n = 32)	81% (n = 47)	0.892
anticoagulatory long-term medication	22.9% (n = 27)	20% (n = 4)	22.5% (n = 9)	24.1% (n = 14)	0.928

^1^ The demographic data as well as the injury severity of the body regions represented by the Abbreviated Injury Scale (AIS) and the Injury Severity Score (ISS) respectively New Injury Severity Score (NISS) depending on the fall height in stair and escalator falls are presented. Abbreviations: IQR: interquartile range, SD: standard deviation.

**Table 2 jcm-11-00697-t002:** Preclinical and clinical parameters. ^2^

	Total (n = 118)	Low Fall Height (n = 20)	Intermediate Fall Height (n = 40)	High Fall Height (n = 58)	*p*-Value
GCS_at scene_ (points), mean ± SD	11.7 ± 4.3	11.6 ± 4.7	11.9 ± 4.4	11.7 ± 4.2	0.972
GCS_in-hospital_ (points), mean ± SD	12.1 ± 4.7	11.9 ± 5.1	12.6 ± 4.3	11.7 ± 4.8	0.693
preclinical intubation	15.3% (n = 18)	20% (n = 4)	10% (n = 4)	17.2% (n = 10)	0.454
SBP_in-hospital_ (mmHg), mean ± SD	151.6 ± 35.8	150 ± 27.4	160.5 ± 36.2	146.1 ± 37.5	0.160
HR_in-hospital_ (bpm), mean ± SD	85.1 ± 18.9	80.4 ± 17.4	85.9 ± 22.1	86.2 ± 16.9	0.495
hemoglobin _in-hospital_ (g/dL), mean ± SD	13.1 ± 1.9	12.9 ± 1.5	13.3 ± 1.8	13.1 ± 2.1	0.691
platelet count _in-hospital_ (cells/µL), mean ± SD	208,263.9 ± 70,446.1	192,944.4 ± 53,932.9	200,062.2 ± 76,988.9	218,607.1 ± 70,110.9	0.280
positive BAC	44.9% (n = 53)	25% (n = 5)	47.5% (n = 19)	50% (n = 29)	0.141

^2^ Shown are preclinical and clinical parameters as a function of the number of fallen steps. Abbreviations: BAC: blood alcohol concentration, bpm: beats per minute, dL: deciliter g: gram, GCS: Glasgow Coma Scale, HR: heart rate, µL: microliter, mmHg: millimeters of mercury, SBP: systolic blood pressure, SD: standard deviation.

**Table 3 jcm-11-00697-t003:** Most frequent consequences of injuries. ^3^

	Total (n = 118)	Low Fall Height (n = 20)	Intermediate Fall Height (n = 40)	High Fall Height (n = 58)	*p*-Value
TBI	83.1% (n = 98)	75% (n = 15)	80% (n = 32)	87.9% (n = 51)	0.338
EDH	4.2% (n = 5)	0% (n = 0)	0% (n = 0)	9.4% (n = 5)	0.067
SDH	19.5% (n = 23)	5% (n = 1)	12.5% (n = 5)	29.3% (n = 17)	0.024
SAH	24.6% (n = 29)	5% (n = 1)	25% (n = 10)	31% (n = 18)	0.066
ICH	19.5% (n = 23)	5% (n = 1)	15% (n = 6)	27.6% (n = 16)	0.06
fractures	47.6% (n = 56)	10% (n = 2)	35% (n = 14)	68.9% (n = 40)	0.015

^3^ The injury consequences are shown depending on the height of the fall. Most frequently, patients suffered a traumatic brain injury (TBI), which was further differentiated into the four bleeding entities epidural hemorrhage (EDH), subdural hemorrhage (SDH), subarachnoidal hemorrhage (SAH) and intracerebral hemorrhage (ICH).

**Table 4 jcm-11-00697-t004:** Outcome. ^4^

	Total (n = 118)	Low Fall Height (n = 20)	Intermediate Fall Height (n = 40)	High Fall Height (n = 58)	*p*-Value
emergency operation head	10.2% (n = 12)	5% (n = 1)	10% (n = 4)	12.1% (n = 7)	0.652
operation (head excluded)	24.6% (n = 29)	10% (n = 2)	30% (n = 12)	25.9% (n = 15)	0.947
LOS ICU (days), mean ± SD	2.9 ± 6.4	1.3 ± 2.8	2.2 ± 3.8	4.1 ± 8.4	0.156
LOS hospital (days), mean ± SD	7.4 ± 9.1	4.8 ± 4.4	7.5 ± 9.2	8.3 ± 9.9	0.334
mortality	11% (n = 13)	10% (n = 2)	10% (n = 4)	12.1% (n = 7)	0.938

^4^ Shown are outcome parameters in relation to the number of stairs fallen. Abbreviations: ICU: intensive care unit, LOS: length of stay, SD: standard deviation.

**Table 5 jcm-11-00697-t005:** Comparison between patients under the influence of alcohol and patients not under the influence of alcohol. ^5^

	BAC Negative	BAC Positive	*p*-Value
BAC_admission_ (g/dL), mean ± SD	0	2.45 ± 0.98	0.000
BAC_admission_ (per mille), mean ± SD	0	1.99 ± 0.79	0.000
fall height (number of steps), mean ± SD	10.8 ± 5.6	11.9 ± 5.5	0.219
GCS_at scene_ (points), mean ± SD	11.5 ± 4.7	11.9 ± 3.9	0.499
GCS_in-hospital_ (points), mean ± SD	11.4 ± 5.2	12.8 ± 3.9	0.624
ISS (points), median (IQR)	8 (16)	5 (9)	0.195
NISS (points), median (IQR)	9 (20)	6 (12)	0.335
AIS_head_ (points), mean ± SD	2.17 ± 1.58	1.81 ± 1.51	0.205
AIS_face_ (points), mean ± SD	0.32 ± 0.72	0.43 ± 0.81	0.406
AIS_thorax_ (points), mean ± SD	0.77 ± 1.29	0.21 ± 0.74	0.004
AIS_abdomen_ (points), mean ± SD	0.32 ± 0.74	0.36 ± 0.71	0.896
AIS_extremities_ (points), mean ± SD	0.38 ± 0.82	0.23 ± 0.64	0.215
LOS ICU (days), mean ± SD	3.5 ± 7.7	2.3 ± 4.3	0.479
LOS hospital (days), mean ± SD	8.3 ± 10.4	6.3 ± 6.8	0.372

^5^ Presented are preclinical and clinical parameters as well as injury characteristics depending on blood alcohol concentration (BAC). Abbreviations: AIS: Abbreviated Injury Scale, bpm: beats per minute, dL: deciliter g: gram, GCS: Glasgow Coma Scale, ICU: Intensive care unit, IQR: interquartile range, ISS: Injury Severity Score, NISS: New Injury Severity Score, LOS: length of stay, SD: standard deviation.

## Data Availability

Data is contained within the article.

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
