# Peer review of "Injury Patterns after Falling down Stairs—High Ratio of Traumatic Brain Injury under Alcohol Influence"

_jcm, 2022, doi:10.3390/jcm11030697_

Round 1
Reviewer 1 Report
I would like to thank the authors for their simple epidemiological paper on the prevalence of head injuries from falls of differing heights. The message is simple: a large proportion have TBI, irrelevant of the height, and thus should be not be deprived of advanced imaging as clinical examination may be unreliable.
The paper is well constructed and can be accepted in its current for, as the message is well supported by the manuscript content.
Author Response
Dear Prof. Balogh,
Dear Reviewers,
Thank you for reviewing our manuscript “Injury patterns after falling down stairs - an independent category of fall events?”, and for all the valuable comments made by the reviewers. We agree completely with the comments and we have addressed them below. Changes have been made to reflect this, and the revised manuscript with highlighted changes has been attached.
Reviewer #1
I would like to thank the authors for their simple epidemiological paper on the prevalence of head injuries from falls of differing heights. The message is simple: a large proportion have TBI, irrelevant of the height, and thus should be not be deprived of advanced imaging as clinical examination may be unreliable. The paper is well constructed and can be accepted in its current for, as the message is well supported by the manuscript content.
We are very pleased that the paper fully meets the requirements of the first reviewer.
Reviewer 2 Report
Thank you very much for allowing me to review your work. I think the authors address an important issue which is of great importance.
Here are some comments and recommendations about your work:
In my opinion, the biggest problem with this study is that the authors, already from the title, expose that they want to know if the falls down the stairs suppose a different group to other types of falls; but in his research the sample is composed only of subjects who have fallen down the stairs, so no comparison can be made with people who fall in environments without stairs either from low or high height. For this reason, I believe that the statements that are made throughout the document in relation to it should be eliminated. This could be posed as a research question for future derivative studies.
Lines 37-43: Are there any publications that support these claims? If there is not, these types of statements in the introduction do not proceed, since it is about what the authors are investigating (it is the question posed in the title), so it cannot be said that this is so before the presentation of their research. If your data agrees with this research, they could make these claims in the discussion and results, but, in my opinion, in the introduction, it is not appropriate.
Lines 68-72: Is there any literature that supports this division of the number of steps or is it a division made by researchers? What are they based on for this?
What are the hypotheses from which your study starts in relation to the objectives set? There is information that is repeated in text and tables; the information is only necessary to provide it once either in the text or in the table.
Author Response
Dear Prof. Balogh,
Dear Reviewers,
Thank you for reviewing our manuscript “Injury patterns after falling down stairs - an independent category of fall events?”, and for all the valuable comments made by the reviewers. We agree completely with the comments and we have addressed them below. Changes have been made to reflect this, and the revised manuscript with highlighted changes has been attached.
Reviewer #2
Thank you very much for allowing me to review your work. I think the authors address an important issue which is of great importance.
Here are some comments and recommendations about your work:
In my opinion, the biggest problem with this study is that the authors, already from the title, expose that they want to know if the falls down the stairs suppose a different group to other types of falls; but in his research the sample is composed only of subjects who have fallen down the stairs, so no comparison can be made with people who fall in environments without stairs either from low or high height. For this reason, I believe that the statements that are made throughout the document in relation to it should be eliminated. This could be posed as a research question for future derivative studies.
Thank you for this important comment. We understand that the choice of the title may suggest that in the present study stair falls could be compared with another type of fall entity, such as ground level falls or falls from greater heights. The primary objective of this work was to analyze the pattern of injury after stair falls as a function of the number of stairs fallen. Accordingly, we adapted the title as follows: "Injury patterns after falling down stairs - high ratio of traumatic brain injury under alcohol influence". Consequently, further text passages in the paper have also been adjusted.
Lines 37-43: Are there any publications that support these claims? If there is not, these types of statements in the introduction do not proceed, since it is about what the authors are investigating (it is the question posed in the title), so it cannot be said that this is so before the presentation of their research. If your data agrees with this research, they could make these claims in the discussion and results, but, in my opinion, in the introduction, it is not appropriate.
We agree with your point of view that the mentioned statements are not correctly placed in the introduction and therefore they have been deleted from the introduction section. However, in our opinion, it is very important to categorize stair falls as a relevant subset of falls and to explicitly place the resulting injury patterns in discourse with the existing data in the literature of immediate injury consequences of ground level falls and falls from greater heights. Therefore, this paragraph has been added to the discussion section.
Lines 68-72: Is there any literature that supports this division of the number of steps or is it a division made by researchers? What are they based on for this?
To our knowledge, there was no explicit division of fall height in the previously published studies on stair falls. While there are some publications that examine injury outcomes after a stair fall, none describe the direct relationship of injury pattern to the number of stairs fallen. Therefore, the selected division is freely chosen by us. We think that with this division appropriate cohorts could be formed, which allowed a reasonable comparability.
What are the hypotheses from which your study starts in relation to the objectives set? There is information that is repeated in text and tables; the information is only necessary to provide it once either in the text or in the table.
Thank you for this useful advice. You are absolutely right that it is quite enough if the detailed information is presented either in the text or in the tables. Therefore, the redundant information has been deleted in the current version.
Round 2
Reviewer 2 Report
The authors have complied with my suggestions.